# Metabolite Profiling of the Gut–Renal–Cerebral Axis Reveals a Particular Pattern in Early Diabetic Kidney Disease in T2DM Patients

**DOI:** 10.3390/ijms24076212

**Published:** 2023-03-25

**Authors:** Lavinia Balint, Carmen Socaciu, Andreea Iulia Socaciu, Adrian Vlad, Florica Gadalean, Flaviu Bob, Oana Milas, Octavian Marius Cretu, Anca Suteanu-Simulescu, Mihaela Glavan, Silvia Ienciu, Maria Mogos, Dragos Catalin Jianu, Ligia Petrica

**Affiliations:** 1Department of Internal Medicine II—Division of Nephrology, “Victor Babes” University of Medicine and Pharmacy Timisoara, County Emergency Hospital Timisoara, Eftimie Murgu Sq. No. 2, 300041 Timisoara, Romania; 2Center for Molecular Research in Nephrology and Vascular Disease, Faculty of Medicine, “Victor Babes” University of Medicine and Pharmacy, Eftimie Murgu Sq. No. 2, 300041 Timisoara, Romania; 3Research Center for Applied Biotechnology and Molecular Therapy Biodiatech, SC Proplanta, Trifoiului 12G, 400478 Cluj-Napoca, Romania; 4Department of Occupational Health, University of Medicine and Pharmacy “Iuliu Haţieganu”, Victor Babes 8, 400347 Cluj-Napoca, Romania; 5Department of Internal Medicine II—Division of Diabetes and Metabolic Diseases, “Victor Babes” University of Medicine and Pharmacy Timisoara, County Emergency Hospital Timisoara, Eftimie Murgu Sq. No. 2, 300041 Timisoara, Romania; 6Department of Surgery I—Division of Surgical Semiology I, “Victor Babes” University of Medicine and Pharmacy Timisoara, Emergency Clinical Municipal Hospital Timisoara, Eftimie Murgu Sq. No. 2, 300041 Timisoara, Romania; 7Department of Neurosciences—Division of Neurology, “Victor Babes” University of Medicine and Pharmacy Timisoara, County Emergency Hospital Timisoara, Eftimie Murgu Sq. No. 2, 300041 Timisoara, Romania; 8Center for Cognitive Research in Neuropsychiatric Pathology (Neuropsy-Cog), Faculty of Medicine, “Victor Babes” University of Medicine and Pharmacy, Eftimie Murgu Sq. No. 2, 300041 Timisoara, Romania; 9Center for Translational Research and Systems Medicine, Faculty of Medicine, “Victor Babes” University of Medicine and Pharmacy, Eftimie Murgu Sq. No. 2, 300041 Timisoara, Romania

**Keywords:** diabetic kidney disease, gut microbiota-derived biomarkers, nitrogen metabolic pathway, retinoic acid signaling pathway, blood–brain barrier, podocyte injury, proximal tubule dysfunction

## Abstract

Type 2 diabetes mellitus (T2DM) represents an important microvascular disease concerning the kidney and the brain. Gut dysbiosis and microbiota-derived metabolites may be in relation with early pathophysiological changes in diabetic kidney disease (DKD). The aim of the study was to find new potential gut-derived biomarkers involved in the pathogenesis of early DKD, with a focus on the complex interconnection of these biomarkers with podocyte injury, proximal tubule dysfunction, renal and cerebrovascular endothelial dysfunction. The study design consisted of metabolite profiling of serum and urine of 90 T2DM patients (subgroups P1-normoalbuminuria, P2-microalbuminuria, P3-macroalbuminuria) and 20 healthy controls (group C), based on ultra-high-performance liquid chromatography coupled with electrospray ionization-quadrupole-time of flight-mass spectrometry analysis (UHPLC-QTOF-ESI^+^-MS). By multivariate and univariate analyses of serum and urine, which included Partial Least Squares Discriminant Analysis (PLSDA), Variable Importance Plots (VIP), Random Forest scores, One Way ANOVA and Biomarker analysis, there were discovered metabolites belonging to nitrogen metabolic pathway and retinoic acid signaling pathway which differentiate P1 group from P2, P3, C groups. Tyrosine, phenylalanine, indoxyl sulfate, serotonin sulfate, and *all-trans* retinoic acid express the metabolic fingerprint of P1 group vs. P2, P3, C groups, revealing a particular pattern in early DKD in T2DM patients.

## 1. Introduction

Diabetic kidney disease (DKD) represents a healthcare issue by affecting approximately 40% of patients suffering from diabetes mellitus (DM). In spite of signs of abating, the incidence of DKD continues to rise worldwide, and is the main cause of end-stage renal disease (ESRD). It is well known that DKD is triggered by uncontrolled hyperglycemia, through several mechanisms such as hemodynamic, metabolic, inflammatory and epigenetic disturbances leading to renal fibrosis and ESRD [1,2,3,4].

Endothelial dysfunction is a systemic epiphenomenon, and is characteristic to T2DM as a result of a long-standing hyperglycemic state. A dysregulated glycemic milieu triggers the production of reactive oxygen species (ROS), which determine nitric oxide (NO) imbalances and an increased release of chemokines [monocyte chemoattractant protein-1—(CCL-2), chemokine (C-C motif) ligand 5 (CCL-5)], cytokines [(tumor growth factor β (TGF-β), tumor growth factor α (TGF-α), interleukin 6 (IL-6) and interleukin 8 (IL-8)] and adhesion molecules [(E-selectin and vascular cell adhesion molecule (VCAM)]. In the end, endothelial dysfunction manifests as an inflammatory state, characterized by inadequate vasodilatation, with a subsequent loss of the endothelial cell function of a barrier and proliferation capacity [5].

Besides renal complications, DKD leads to cardiovascular and cerebrovascular complications, making the management of the disease even more challenging. A cerebral microvascular disease secondary to T2DM manifests as a disruption of the blood–brain barrier (BBB) homeostasis, which is mainly composed of endothelial cells, astrocytes and pericytes. These cells are unable to uptake high quantities of glucose, triggering an inflammatory process which weakens endothelial cell tight junctions and determines an increased permeability through the BBB. Ultimately, cerebral microvascular disease may lead to lacunar polytopic infarction, depression and dementia [6,7]. 

In the last decade, more studies recognize the outstanding role of intestinal dysbiosis and gut-derived metabolites in the development of DKD [8,9]. A new spectrum of research regarding the pathogenesis of DKD is represented by the gut–renal axis. 

Gut microbiota consists of trillions of bacteria that have a symbiotic connection with the host. Prolonged uremia may cause an imbalance in the gut homeostasis, followed by dysbiosis and a high permeability through the intestinal barrier [10,11]. Therefore, the metabolites resulted in the gut are absorbed into the bloodstream and may have unfavorable effects by producing endothelial dysfunction in organs with similar structures, such as the small vessels of the brain and the kidney [12,13]. In parallel, these metabolites may be involved in podocyte injury and proximal tubular dysfunction by acting on specific receptors such as aryl hydrocarbon receptor (AHR) and organic anion transporters (OATs) [14].

Metabolomics and metabolite profiling/fingerprinting offer a comprehensive measurement of metabolic changes, from a cellular to systemic level, as a response to different pathologies and intrinsic or extrinsic agents, including diet, gut microbiota and lifestyle. Metabolomics represents a new field of interest and a systematic approach to biomarker discovery, such as small molecules (less than 5000 Da), as metabolites related to cellular metabolism. It includes the understanding of certain metabolic pathways and the identification of gut microbiota metabolites with a possible role in the early diagnosis, prevention and treatment of DKD [15]. 

Metabolomics includes two types of approaches, such as the untargeted profiling and the targeted analysis of certain metabolites. The untargeted profiling of metabolites allows the simultaneous separation and identification of many metabolites using mostly the advanced liquid chromatography/mass spectrometry (LC/MS)-based techniques and provides their general fingerprint, based on retention time and mass/charge ratios, unlike the targeted metabolomics, where a defined set of metabolites is analyzed and quantified [16]. This study applied the untargeted metabolite profiling in the serum and urine of T2DM patients UHPLC-QTOF-ESI+-MS Analysis, combined with a statistical multivariate and univariate analyses. 

As DKD incidence is continuously rising, metabolomic techniques are promising tools for achieving specific biomarkers of incipient DKD. Hence, new therapeutical strategies may be developed, as the patients’ care, based on the individualized metabolomic profile, is becoming the trending topic of modern medicine. The aim of this study was to find new potential gut-derived biomarkers, with a focus on the complex interconnection of these metabolites with podocyte injury, proximal tubular dysfunction, and renal and cerebrovascular endothelial dysfunction, which are all involved in the pathogenesis of early DKD. 

## 2. Results

According the raw data obtained by the procedures mentioned in the materials and methods (Section 4.3 and Section 4.4), the final number of molecules selected for statistical analysis was 136 in serum and 196 in urine.

### 2.1. The Comparison of Patients with DKD (P Group) and the Control Individuals (C Group) by Multivariate Analysis of Serum and Urine Samples

Serum metabolites were further classified based on a PLSDA analysis, and a significant difference between groups was observed in the score plot (Figure 1a), with a covariance of 13%. In addition, the VIP scores based on PLSDA results revealed the first 15 molecules to be considered as responsible for the discrimination between T2DM patients (group P) and healthy control subjects (group C). 

The molecules with VIP values >2 were identified: *m*/*z* values of 182.0932 (tyrosine) 188.0831 (N1-Acetylspermidine), 177.0663 (Threonylglycine), 190.0625 (Kynurenic acid) 158.1648 (Tiglylglycine), 340.2803 (Oleoylglycine), 203.0661 (Dimethylarginine) 301.1598 (*all-trans* retinoic acid) and 156.0864 (Histidine). Excepting tyrosine, acetylspermidine, kynurenic acid and histidine, all the other molecules had increased values in the P group. The cross-validation plot presented a high accuracy, with high R2 values (>0.8) and a significant Q2 value (>0.6) for the first component, confirming a good predictability of the model. 

Moreover, the prediction of the potential biomarkers was achieved by a Random Forest analysis, with 15 molecules identified as putative biomarkers based on the Mean Decrease Accuracy (MDA) values (Figure 1b). Out of these molecules, kynurenic acid, oleoylglycine, tyrosine and dimethylarginine were also found and may be considered as putative biomarkers of differentiation. 

Subsequently, the biomarker analysis was represented by the Receiver Operating Characteristic (ROC) curves and the Area Under Curve (AUC). There were revealed 19 molecules with AUC values higher than 0.7. In accordance with the previous analysis, the biomarker analysis displayed four molecules in the serum with AUC values higher than 0.8: All-trans retinoic acid, Cysteine-S-sulfate, Threonylglycine and Dimethylarginine were considered as discriminative between group P vs. group C.

The urine metabolites were categorized and identified by the same methods, and the statistical analysis was conducted similarly by applying the PLSDA analysis (Figure 2a), including the VIP scores. Moreover, in this case, a good discrimination between groups C and P was obtained, and the VIP scores revealed the 15 molecules that may distinguish group P from group C. The molecules with VIP scores >2.4 had *m*/*z* values of 275.1649 (serotonin sulfate), 214.255 (Indoxyl sulfate), 200.2356 (O-Phosphothreonine), 279.1637 (Leucyl-phenylalanine), 178.0609 (N-Formyl-L-methionine) and 251.0669 (Methionyl-threonine). The cross-validation plot showed an acceptable accuracy of results with R2 values higher than 0.5, but negative Q values. 

Additionally, the Random Forest analysis revealed 15 molecules with a potential role in DKD with MDA scores > 0.01 (Figure 2b). All these molecules had higher levels in group P. These molecules were categorized by a biomarker analysis, with five molecules generating interest based on AUC score values higher than 0.9. These molecules were represented by serotonin sulfate, Indoxyl sulfate, *all-trans* retinoic acid, Glycylprolylarginine and Prostaglandin E2.

### 2.2. The Comparison of DKD Subgroups (Normoalbuminuria-P1, Microalbuminuria-P2 and Macroalbuminuria-P3) and Control Group (C) in Serum and Urine

The univariate serum and urine sample analysis allowed for the classification of the molecules identified by the multivariate analysis in the three subgroups (P1, P2, P3), in order to identify the molecular pattern of the incipient stage of DKD (P1 subgroup) vs. P2, P3 subgroups and C group.

By applying the One-Way ANOVA analysis and Fischer’s LSD correlation, 20 serum molecules were found to mark differences between subgroups. The most significant molecules identified, based on *m*/*z* values, were: 216.0103 (Propeonyl carnitine), 340.2809 (Oleoyl glycine), 179.0152 (Cysteinyl glycine), 301.1610 (*all-trans* retinoic acid) and 169.1529 (Pyridoxamine)

For the further classification, the Hunter Pattern analysis was performed with 16 serum molecules being identified. The most significant molecules (*p* < 0.05) based on *m*/*z* retention times were: 177.0663 (Threonylglycine), 103.0527 (Acetoacetic acid), 249.2229 (Glutamylthreonine), 191.1770 (5-Methoxymethyl adipic acid), 190.0625 (Kynurenic acid), 166.0975 (Phenylalanine) and 301.1598 (*all-trans* retinoic acid). The cross-validation plot showed an acceptable accuracy of results with R2 values higher than 0.1.

Subsequently, the PLSDA score (Figure 3a), based on VIP values, revealed 15 serum molecules that express a good discrimination between the subgroups and C group, including 190.0625 (Kynurenic acid), 166.0975 (Phenylalanine) and 301.1598 (*all-trans* retinoic acid). In addition, the Random Forest analysis revealed 15 molecules as putative biomarkers based on MDA values (Figure 3b).

The univariate analysis of urine samples was performed similarly to serum samples. By applying One-Way ANOVA and Hunter pattern analyses, six molecules demonstrated a significant differentiation between subgroups: 275.1649 (serotonin sulfate), 214.2550 (Indoxyl sulfate), 200.2356 (O-Phosphothreonine), 178.0609 (N-Formyl-L-methionine), 189.1594 (Cresyl sulfate) and 188.0709 (N1-Acetylspermidine).

PLSDA score plots revealed 15 molecules in urine that are responsible for the discrimination of subgroups (Figure 4a). The cross-validation plot showed an acceptable accuracy of results, with R2 values higher than 0.1. The prediction of the potential biomarkers was achieved by the Random Forest analysis, with 15 molecules being identified as putative biomarkers based on MDA values (Figure 4b). Out of these molecules, Indoxyl sulfate, *all-trans* retinoic acid, serotonin sulfate, Glycylprolylarginine and O-Phosphothreonine were also found and may be considered as key molecules in urine. In the end, the determination of urinary molecules was followed by their correlation with urinary creatinine for obtaining more accurate results.

### 2.3. Integration of Statistical Results

The multiple data released from the LC-MS analysis coupled with multivariate and univariate statistics were compared and integrated in order to select the potential metabolites which may be considered as representative to discriminate between the stages of the DKD and C group. Table 1a,b summarizes the potential metabolites and their variations in these groups in serum (Table 1a) and urine (Table 1b).

Moreover, the classification of metabolites was achieved by their division into classes. A semi-targeted analysis was performed in order to point out the dynamics of these molecules in the P1, P2 and P3 subgroups and in group C according to their peak intensity. Based on this final classification, it was observed that the nitrogen metabolic pathway (phenylalanine, tyrosine and tryptophan metabolites—indoxyl sulfate and serotonin sulfate), and retinoic acid signaling pathway (*all-trans* retinoic acid) may highlight a particular pattern by the comparison of the P1 subgroup with the P2, P3 and C group. 

In serum, there were selected, as potential biomarkers, *all-trans* retinoic acid, phenylalanine, treonylglycine, oleoylglycine and other amino-acid derivatives (dimethylarginine, methoxytryptophan and glutamylthreonine). 

In urine, all-trans retinoic acid, indoxyl sulfate, O-phosphothreonine, leucylphenylalanine and serotonin sulfate had similar variations, as they were increased in the subgroup P1, with the maximum levels in the P3 subgroup. Meanwhile, p-cresol sulfate and 5-Hydroxy lysine had opposite variations, as they were reduced in P1–P3 groups. 

The variations of these metabolites suggest the involvement of the amino acid-related metabolic pathways in serum and urine, e.g., phenylalanine-tyrosine-tryptophan-kinurenic acid pathways, including their metabolites (e.g., acylated glycine, leucyl phenylalanine, methionyl threonine, phosphothreonine and serotonin sulfate). 

In order to find out the potential implication of these metabolites in early DKD (their trending) between subgroups was studied (P1 vs. P2, P1 vs. P3 and P1 vs. C). Moreover, their final selection was guided by data found in literature about their gut provenance and their implications in BBB dysfunction. It was concluded that phenylalanine, tyrosine, indoxyl sulfate, serotonin sulfate and all-trans retinoic acid may be potential serum biomarkers of incipient DKD and endothelial dysfunction. In parallel, phenylalanine, indoxyl sulfate, serotonin sulfate and *all-trans* retinoic acid may express the biomarker potential in urine. 

## 3. Discussion

This study was conducted on T2DM patients, in order to find new potential gut-derived biomarkers involved in the pathogenesis of early DKD, based on the UHPLC-QTOF-ESI^+^-MS analysis. By multivariate and univariate analyses of serum and urine, metabolites belonging to the nitrogen metabolic pathway and retinoic acid signaling pathway were discovered that differentiate T2DM patients from controls. Tyrosine, phenylalanine, tryptophan, indoxyl sulfate, serotonin sulfate and *all-trans* retinoic acid may be considered as key candidate molecular markers, as well as potential therapeutical targets in early DKD.

### 3.1. Phenylalanine and Tyrosine Metabolism May Be Involved in Early DKD

In our study, decreased levels of phenylalanine and tyrosine were observed in the DKD vs. C group in the serum and urine. These data are in good agreement with previous publications which reported lower urinary tyrosine in T2DM micro- to macroalbuminuria cases versus controls [17]. Furthermore, the individual comparison of the subgroup P1 vs. P2 and P1 vs. P3, pointed out that the normoalbuminuric subgroup expresses higher levels of phenylalanine both in serum and urine. In addition, tyrosine followed the same pattern as phenylalanine. 

Phenylalanine and tyrosine are dietary aromatic amino acids that are metabolized in the gut and involved in the nitrogen metabolic pathway. Their dynamics are codependent, as tyrosine is synthetized from phenylalanine, with phenylalanine hydroxylase (PAH) as a substrate. Interestingly, the activity of PAH is highly influenced by tetrahydrobiopterin (BH4), a cofactor also involved in nitric oxide synthesis, which is responsible for modulating the endothelial cell function [18].

According to Rygula et al., disturbances in the phenylalanine and tyrosine ratio are a consequence of impaired PAH activity due to the BH4-reduced bioavailability. Decreased BH4 activity determines imbalances in NO production, with subsequent endothelial dysfunction and impairment in neurotransmission [19]. Hence, our results may suggest the implication of phenylalanine and tyrosine in incipient renal and cerebrovascular endothelial dysfunction due to BH4 impairment, when comparing the P1 vs. P2 and P1 vs. P3 subgroups. 

Moreover, it was recently demonstrated that low phenylalanine and tyrosine work synergistically and interact with diabetic nephropathy to increase the risk of diabetic retinopathy [20]. This dynamic interconnection between phenylalanine and tyrosine levels was also reported to be altered in early stages of chronic kidney disease, as well as in type 2 DM [21,22]. In addition, the disturbances in the gut’s nitrogen metabolism seem to occur years before the diagnosis of DM, with the phenylalanine pathway being involved in the early pathogenic mechanism of DM development [23]. Focused research on these changes may provide additional information with regard to new biomarkers of early DKD related to the nitrogen metabolic pathway.

### 3.2. Tryptophan Metabolites May Be Involved in the Cross-Talk of Podocyte Injury, Proximal Tubule Dysfunction and Endothelial Dysfunction 

Indoxyl sulfate, a gut metabolite and a hallmark of tryptophan metabolism, may behave as a uremic toxin or as a cell-regulator [24]. In our study, high levels of indoxyl sulfate were observed in the DKD group vs. the control group, with higher values in urine vs. blood, according to the multivariate analysis. Furthermore, the univariate serum analysis revealed a progressive increase in indoxyl sulfate levels, in concordance with the DKD progression (P1 < P2 < P3). By the urine univariate analysis, levels of indoxyl sulfate were reported to be discriminative of the early phases of DKD, namely in the normoalbuminuric group when compared with the C group and P2 and P3 subgroups (C < P1 < P2 < P3).

Indoxyl sulfate seems to be central to renal damage through mechanisms that involve the glomeruli and tubules, as well as the endothelium. The dynamics of indoxyl sulfate in relation to the progressive decline of the glomerular filtration rate implies the activation of the aryl hydrocarbon receptor, which downregulates podocyte proteins involved in cell integrity, thus leading to podocyte foot processes’ effacement and proteinuria [25]. Moreover, according to Lowenstein et al., indoxyl sulfate levels are rising in blood in the context of renal impairment, the excretion of which occurs in the proximal tubule through OATs [26].

Given the fact that the early stage of DKD is characterized by hyperfiltration and the absence of albuminuria, and due to the protein-bound structure of indoxyl sulfate, we may speculate that the dysregulated urinary excretion of this metabolite in the normoalbuminuric subgroup vs. control group may reflect a renal tubular damage that precedes glomerular modifications in DKD. These findings are indicative of the fact that indoxyl sulfate may be a potential biomarker of early proximal tubular damage in DKD pathogenesis, even before the occurrence of albuminuria. In addition, the indoxyl sulfate’s accumulation in serum may determine the endothelial cell damage through the increased production of reactive oxygen species [27]. The report by Gao et al. includes fascinating features concerning OATs that also seem to be expressed by the endothelium of cerebral small vessels, which are a part of the blood–brain barrier [28]. Moreover, studies performed by Bobot et al. on mice with chronic kidney disease revealed the potential involvement of indoxyl sulfate in BBB dysfunction, a process mediated by pro-inflammatory cytokines such as TNF-α and IL-6 [29]. Besides podocyte and tubular damage, indoxyl sulfate might as well be a central player in the gut–renal–cerebral axis in terms of the endothelial dysfunction through OAT receptors and pro-inflammatory cytokines. However, further studies are needed in order to establish this interorgan connection in DKD.

Moreover, our study displayed high levels of serotonin sulfate by the comparison of P1 vs. C and P2 and lower levels of P1 vs. P3 in serum. In parallel, the urine analysis revealed higher levels of serotonin sulfate in the P1 subgroup when compared with the other subgroups. Serotonin, a molecule involved in neurotransmission and vascular tone, is a derivative of tryptophan metabolism. Recent studies point out its possible involvement in the incipient renal damage in diabetic patients by triggering the transforming growth factor β (TGF-β) mediated mesangial cell proliferation with subsequent renal fibrosis [30]. The serotonin-sulfate results from serotonin sulfation by the gut bacteria. It is a lesser known molecule in terms of cerebral and renal damage and seems to have an interesting trend in our study. Serotonin sulfate deserves more attention in further studies, as its levels are discriminative between the normoalbuminuric group and the other groups.

### 3.3. Retinoic Acid Signaling Pathway Could Have Beneficial Effects on Renal Structures and on Cerebrovascular Endothelium 

Our results characterize the retinoic acid signaling pathway as a possible important contributor to the pathogenesis of DKD. Based on these results, the *all-trans* retinoic acid in serum is progressively increasing from the C to P1-P2-P3 subgroup. In urine, the comparison of P1 vs. C, P2 and P3 revealed higher levels in the normoalbuminuric group.

It is well known that retinoic acid (RA) is the dominant analogue derived from dietary vitamin A (retinol). This exerts its functions through six biologically active metabolites, *all-trans* retinoic acid being the dominant one [31]. Partially produced in the small intestine and flowing through the bloodstream in association with serum albumin, the concentration of *all-trans* retinoic acid dictates the well-functioning of the retinoic acid signaling pathway. The point of action of *all-trans* retinoic acid takes place inside the cell, where it binds to retinoic acid nuclear receptors (RARs), thus upregulating the gene expression and cell differentiation, proliferation, morphogenesis and apoptosis [32]. 

At first sight, the implication of *all-trans* retinoic acid in different metabolisms seems controversial, with beneficial as well as harmful effects. Rhee et al. [24] described an interesting interplay of *all-trans* retinoic acid in DM; namely, this induces pancreatic development and stimulates insulin production in pancreatic islet B cells. On the other hand, the *all-trans* retinoic acid metabolism in DM seems to be profoundly dysregulated by the loss of RAR receptors expressed both in podocytes and in proximal tubular cells. Consequently, “the harmful effects” of *all-trans* retinoic acid are related to the its inability to exert its functions through its receptors because of their low availability [31,33]. 

The RA signaling pathway acts inside the kidney in a two-faced manner. This has an important role in podocyte regeneration by increasing the proliferation of renal progenitor cells. Moreover, in a dysregulated diabetic status, albuminuria binds to RA, the latter being removed through the urine. Thus, the podocytes are deprived of the beneficial effects of RA, with this process leading to podocyte apoptosis and the worsening of albuminuria, triggering a vicious cycle [33,34]. 

From a renal tubular perspective, Molina-Jijon et al. reported in a study performed on mice that the *all-trans* retinoic acid administration improves proteinuria and natriuresis in the diabetic model. Renal tight junction proteins have the claudins as main components, which can be both barrier or pore proteins. Claudin-2 is expressed by renal proximal tubules, while Claudin-5 is expressed by the podocytes, their levels being downregulated in DM and permitting the leakage of sodium and proteins through the urine. By *all-trans* retinoic acid administration, it was demonstrated that claudins’ expression might be upregulated, thus ameliorating the protein and sodium loss [35]. In another study, Sierra-Mondragon et al. reported the important role of *all-trans* retinoic acid in reducing the TGF-β1 overexpression, with the subsequent blockage of renal fibrotic processes in the early stages of DM. It was also reported that *all-trans* retinoic acid upregulates the markers of the proximal tubule injury [Kidney injury molecule-1 (KIM-1) and neutrophil gelatinase-associated lipocalin (NGAL)] in early DKD [36]. 

In a study conducted by Zhang et al. on mice, by common carotid artery ligation, it was demonstrated that *all-trans* retinoic acid administration acts on the activated protein kinase (AMPK) and reduces the proliferation of vascular smooth muscle cells, a process that is dose-dependent. Surprisingly, *all-trans* retinoic acid does the exact opposite to endothelial cells by increasing their capability to multiply [37]. In addition, it was demonstrated that the RA pathway is deeply involved in the formation of vascular and neurological BBB structures. It seems that the administration of *all-trans* retinoic acid modulates the matrix metalloproteinase-9 (MMP-9) expression which is responsible for the BBB remodeling [38].

We speculate that in our P1 subgroup, the *all-trans* retinoic acid levels may be higher in urine as compared to those found in serum, and higher in the P1 vs. C group and P2 and P3 subgroups, as a consequence of a dysregulated diabetic milieu, which may trigger several compensatory processes. First, it may stimulate the pancreas to produce insulin. Second, from a renal viewpoint, *all-trans* retinoic acid may have multiple implications. This may be produced by compensatory into the urinary space in order to stimulate renal progenitor cells to differentiate into podocytes, combat protein and sodium leakage in the proximal tubule, and cease fibrotic processes initiated through the TGF-β1 in early DKD. Its high levels may also be correlated with endothelial dysfunction, as a compensatory mechanism in order to stimulate endothelial cell proliferation, and reestablish endothelial integrity and neuronal recovery. These results strengthen the hypothesis in which the retinoic acid signaling pathway is deeply involved in early DKD pathogenesis.

### 3.4. Clinical Applicability and Potential Therapeutical Aspects

Besides their role in the early DKD diagnosis, nitrogen- and retinoic-acid-derived metabolites may be potential therapeutical targets in the personalized care of DKD patients [39]. Emerging studies now recognize the role of probiotic usage in reducing the gut microbiota proliferation, and are implicit in its metabolite production. Prebiotics, probiotics and symbiotics may represent a reliable option for shifting the elimination of metabolites from a renal to fecal route, in order to interrupt their toxic systemic effects [40]. Furthermore, GLP-1 agonists may need special attention in the management of renal and cerebrovascular metabolite-induced endothelial dysfunction [41].

Our study has several limitations. First, this is a pilot and a cross-sectional study; therefore, the results are observational and do not reflect a relation of causality. Second, we cannot exclude the influence of the glycemic control status, as it may interfere with the comparison between subgroups and be implicit with the statistical power of the study. Third, even though the samples were collected after 8 h of fasting, some patients might have had anterior dietary habits that would eventually interfere with our findings. 

However, our study has its strengths. The data obtained revealed various gut-derived metabolites belonging to the nitrogen metabolic pathway and retinoic acid signaling pathway, which may be associated with damage to renal structures, such as the podocytes and the proximal tubule, and also with the impaired integrity of BBB. In addition, to the best of our knowledge, this is the first study which attempts to describe the gut–renal–cerebral axis in terms of the gut-derived metabolite-mediated endothelial dysfunction in early DKD. 

## 4. Materials and Methods

### 4.1. Patients’ Selection and Compliance with Ethical Standards 

A total number of 130 T2DM patients were screened between July 2021 and April 2022 from the Department and Ambulatory of Nephrology, and from the Department and Ambulatory of Diabetes and Metabolic Diseases, County Emergency Hospital Timisoara. The inclusion criteria consisted of over 5 years of DM duration and a HbA1c less than 10%. All patients were treated with angiotensin 2 converting enzyme inhibitors/angiotensin 2 receptor blockers, statins and oral antidiabetic agents and/or insulin. Patients suffering from uncontrolled DM (HbA1c > 10%), end-stage renal disease, urinary tract infections, other glomerular diseases, autoimmune diseases, infectious diseases, neoplasia, and psychiatric diseases were excluded from the study. Our research was based on a pilot study that consisted of the collection of serum and urine samples from 110 eligible subjects (54 males and 56 females), of which 90 were the T2DM patients (group P) and 20 were healthy control subjects (group C) recruited from the records of general physicians. Group P was divided in three subgroups—normoabuminuria (P1-UACR < 30 mg/g), microalbuminuria (P2-UACR-30-300 mg/g) and macroalbuminuria (P3-UACR > 300 mg/g) (Table 2), according to the KDIGO Guidelines Classification [42]. 

The written informed consent was obtained from all subjects, and the study design was approved by the Ethics Committee for Scientific Research of the “Victor Babes” University of Medicine and Pharmacy Timisoara (29/30.06.2021), as well as by the Ethics Committee of the County Emergency Hospital Timisoara (220/18.01.2021).

### 4.2. Sample Collection and Preparation

Blood was collected by venipuncture in sterile vacutainers without anticoagulant and the serum was stored at −80 °C until the analysis. The samples were labeled using confidential numerical codes. The urine samples were collected from the first morning urine in sterile vials. A volume of 0.8 mL mix of pure HPLC-grade Methanol and Acetonitrile (2:1 *v*/*v*) was added for each volume of 0.2 mL of serum and 0.2 mL of urine, respectively. In each case, the mixture was vortexed to precipitate proteins, ultrasonicated for 5 min and stored for 24 hrs at −20 °C to increase the protein precipitation. The supernatant was collected after centrifugation at 12.500 rpm for 10 min (4 °C) and filtered through Nylon filters (0.2 μm). Finally, the supernatant was placed in glass micro vials and introduced in the autosampler of the ultra-high-performance liquid chromatograph (UHPLC) before injection.

### 4.3. UHPLC-QTOF-ESI^+^-MS Analysis

The metabolomic profiling was performed by ultra-high-performance liquid chromatography coupled with the electrospray ionization-quadrupole-time of flight-mass spectrometry (UHPLC-QTOF-ESI^+^-MS) using a ThermoFisher Scientific (Waltham, MA, USA) UHPLC Ultimate 3000 instrument equipped with a quaternary pump, Dionex delivery system, and MS detection equipment with MaXis Impact (Bruker Daltonics). The metabolites were separated on an Acclaim C18 column (5 μm, 2.1 × 100 mm, pore size of 30 nm (Thermo Scientific) at 28 °C. The mobile phase consisted of 0.1% formic acid in water (A) and 0.1% formic acid in acetonitrile (B). The elution time was set for 20 min. The flow rate was set at 0.3 mL ·min^−1^ for serum samples and at 0.8 mL min^−1^ for urine samples. The gradient for serum samples was 90 to 85% A (0–3 min), 85–50% A (3–6 min), 50–30% (6–8 min), 30–5% (8–12 min), and afterward increased to 90% at 20 min. The gradient for urine samples was 90 to 85% A (0–3 min), 85–30% A (3–6 min) and 30–10% (6–8 min), isocratic until 12 min and then increased until 90% at 20 min. The volume of the injected extract was 5 mL; the column temperature was at 25 °C. Several QC samples obtained from each group were used in parallel to calibrate the separations. Doxorubicin hydrochloride (*m*/*z* = 581.3209) solution 0.5 mg/mL was added in parallel to QC samples as the internal standard. The applied MS parameters were the ionization mode positive (ESI^+^), MS calibration with Natrium formate, capillary voltage—3500 V, pressure for the nebulizing gas—2.8 barr, drying gas flow—12 L/min, and drying temperature—300 °C. The *m*/*z* values to be separated were set between 60 and 600 Daltons. The control of the instrument and the data processing was conducted using the specific software—TofControl 3.2, HyStar 3.2, Data Analysis 4.2 (Bruker, Daltonics, Billerica, MA, USA) and Chromeleon, respectively.

### 4.4. Statistical Analysis

Subsequent to the use of the UHPLC-QTOF-ESI^+^-MS analysis, there were several identified molecules, up to 420 in serum and up to 550 in urine samples, respectively. As a point of entry, the molecules with retention times under 0.8 min, molecules with S/N values < 5, molecules with *m*/*z* above 480 Daltons and minor molecules with peak intensities below 1000 units were removed. Furthermore, the calibration of common molecules (with the same *m*/*z* value) was conducted by using the online software www.bioinformatica.isa.cnr.it (accessed on 8 July 2022) NEAPOLIS [43], with the molecules found in more than 80% of the samples being kept. After using the procedures mentioned above, the final number of molecules was 136 in serum and 196 in urine. These molecules were later introduced in the Metaboanalyst 5.0 platform (https://www.metaboanalyst.ca/, accessed on 8 July 2022) for the analysis (multivariate and univariate). This paper focused on the untargeted, multivariate and univariate analysis of the detected molecules. The comparation of the serum and urine metabolites of the DKD group (P) and the control group (C) was realized by a multivariate analysis. The discrimination between group P and group C was determined by Fold Change, Volcano test, Pattern Hunter analysis, Partial Least Squares Discriminant Analysis (PLSDA), sparse PLSDA and Variable Importance in the Projection (VIP). Moreover, the Random-Forest-based prediction test was applied and the P values were calculated by the *t*-test. The univariate analysis used as analytical methods the One-Way ANOVA, Fischer’s LSD, Pattern Hunter analysis, PLSDA, sparse PLSDA, Random Forest and Biomarker analysis.

## 5. Conclusions

In summary, the metabolomic biomarkers found in our study resulted from phenylalanine, tyrosine, tryptophan and retinoic acid metabolism and revealed a specific metabolic fingerprint in normoalbuminuric type 2 DM patients. Furthermore, the dynamics of these gut-derived metabolites may highlight a common pathogenic pathway within the kidney and the brain with regard to podocyte injury, proximal tubule dysfunction and renal, and cerebral vascular structures. The metabolite profiling of the gut–renal–cerebral axis revealed a particular pattern in early DKD in T2DM patients. Further longitudinal studies, performed by targeted metabolomics, are needed in order to identify gut-derived biomarkers which specifically correlate with renal and cerebrovascular lesions in early DKD.

## Figures and Tables

**Figure 1 ijms-24-06212-f001:**
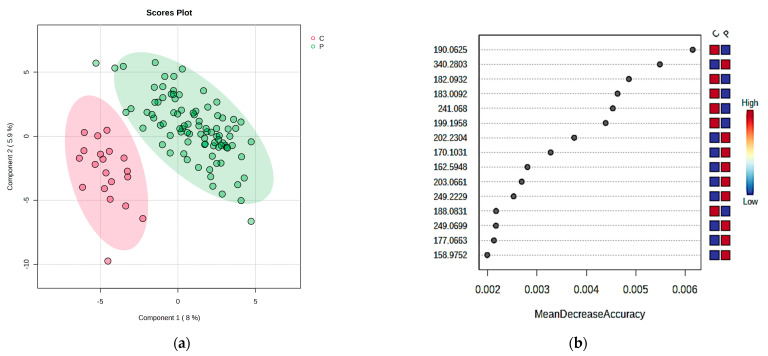
Multivariate analysis of serum samples: (**a**) PLSDA score and plot for serum samples; (**b**) MDA scores according to Random Forest analysis.

**Figure 2 ijms-24-06212-f002:**
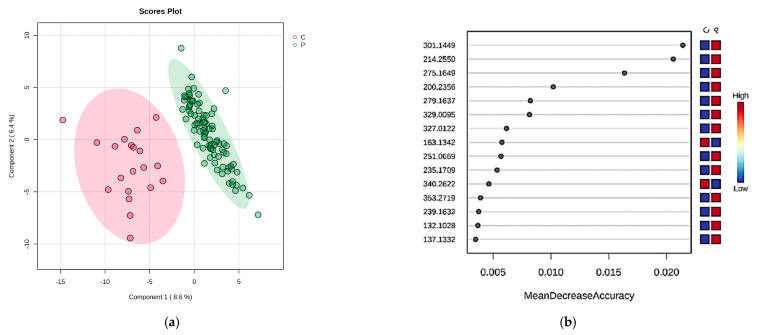
Multivariate analysis of urine samples; (**a**) PLSDA score and plot for urine samples; (**b**) Random Forest: MDA score for urine samples.

**Figure 3 ijms-24-06212-f003:**
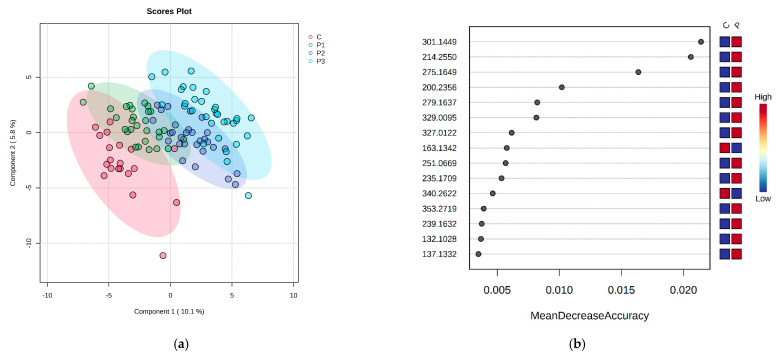
Univariate analysis of serum samples; (**a**) PLSDA score plot of serum samples; (**b**) Random Forest-MDA score of serum samples.

**Figure 4 ijms-24-06212-f004:**
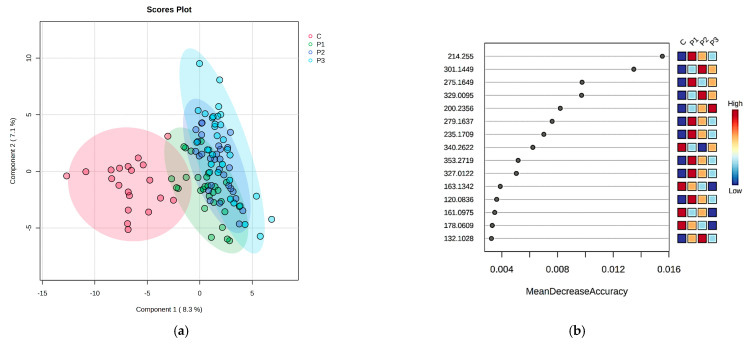
Univariate analysis of urine samples; (**a**) PLSDA score plot of urine samples; (**b**) Random Forest-MDA score of urine samples.

**Table 1 ijms-24-06212-t001:** (**a**) Main serum metabolites to be considered as potential biomarkers of discrimination between healthy and DKD patient group, as such or between the subgroups P1, P2 and P3. The variations, e.g., increases (I) or decreases (D), are also described. (**b**) Main urine metabolites to be considered as potential biomarkers of discrimination between healthy controls (C) vs. DKD group (P), and between healthy controls C vs. subgroups P1, P2 and P3. The variations, e.g., increases (I) or decreases (D), are also mentioned.

**(a)**							
***m*/*z***	**SERUM** **VIP Values > 2 and** **MDA Values > 0.004**	**P vs. C**	***m*/*z***	**SERUM** **VIP Values > 1.6 and** **MDA Values > 0.004**	**P1 vs. P2**	**P1 vs. P3**	**P1 vs. C**
166.0975	Phenylalanine	D	166.0975	Phenylalanine	P1 > P2	P1 > P3	P1 < C
182.0932	Tyrosine	D	182.0932	Tyrosine	P1 > P2	P1 > P3	P1 < C
214.255	Indoxyl sulfate	I	214.255	Indoxyl sulfate	P1 < P2	P1 < P3	P1 < C
190.0625	Kynurenic acid	D	190.0625	Kynurenic acid	P1 > P2	P1 > P3	P1 < C
275.1649	Serotonin sulfate	I	275.1649	Serotonin sulfate	P1 > P2	P1 < P3	P1 > C
301.1598	*all-trans* retinoic acid	I	301.1598	*all-trans* retinoic acid	P1 < P2	P1 < P3	P1 > C
177.0663	Threonylglycine	I	177.0663	Threonylglycine	P1 < P2	P1 > P3	P1 > C
183.0092	Sorbitol	D	183.0092	Sorbitol	P1 > P2	P1 > P3	P1 < C
**(b)**							
***m*/*z***	**URINE** **VIP Values > 2 and** **MDA Values > 0.004**	**P vs. C**	***m*/*z***	**URINE** **VIP Values > 2 and** **MDA Values > 0.004**	**P1 vs. P2**	**P1 vs. P3**	**P1 vs. C**
166.0975	Phenylalanine	D	166.0975	Phenylalanine	P1 < P2	P1 < P3	P1 < C
182.0932	Tyrosine	D	182.0932	Tyrosine	P1 < P2	P1 < P3	P1 < C
214.255	Indoxyl sulfate	I	214.255	Indoxyl sulfate	P1 < P2	P1 < P3	P1 < C
190.0625	Kynurenic acid	D	190.0625	Kynurenic acid	P1 < P2	P1 < P3	P1 < C
275.1649	Serotonin sulfate	I	275.1649	Serotonin sulfate	P1 < P2	P1 < P3	P1 < C
301.1449	*all-trans* retinoic acid	I	301.1449	*all-trans* retinoic acid	P1 < P2	P1 < P3	P1 < C
279.1637	Leucyl-phenylalanine	I	279.1637	Leucyl-phenylalanine	P1~P2	P1~P3	P1 < C
251.0669	Methionyl-threonine	I	189.1594	p-Cresol sulfate	P1 < P2	P1 < P3	P1 < C
329.0095	Glycylprolylarginine	I	163.1342	5-Hydroxy lysine	P1 < P2	P1 < P3	P1 < C

**Table 2 ijms-24-06212-t002:** Demographic and clinical data.

	P1	P2	P3	C
Number of participants	30	30	30	20
Men (nr.,%)	14 (46.66%)	13 (43.33%)	15 (50%)	12 (60%)
Age (y)	68.41 ± 4.98	68.65 ± 4.91	68.84 ± 4.98	55.85 ± 7.25
DM duration (y)	9.6 ± 3.99	9.7 ± 3.99	12.78 ± 3.35	0
Serum creatinine (mg/dL)	0.82 ± 0.18	0.93 ± 0.21	1.07 ± 0.32	0.73 ± 0.08
eGFR (mL/min/1.73 m^2^)	90.42 ± 18.10	89.70 ± 18.19	77.85 ± 19.38	97.93 ± 11.71
UACR (mg/g)	7.38 ± 3.22	45.42 ± 57.08	319.86 ± 585.80	5 ± 0.23
HbA1c (%)	5 ± 0.23	6.42 ± 1.29	7.15 ± 1.60	4.98 ± 0.23

DM—diabetes mellitus; eGFR—estimated glomerular filtration rate; HbA1c—hemoglobin A1c; UACR—urinary albumin/creatinine ratio; data reported as means ± standard deviation

## Data Availability

The data that support the findings of this study are available from the corresponding author upon reasonable request.

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
