# Peer review of "Metabolite Profiling of the Gut–Renal–Cerebral Axis Reveals a Particular Pattern in Early Diabetic Kidney Disease in T2DM Patients"

_ijms, 2023, doi:10.3390/ijms24076212_

Round 1

Reviewer 1 Report (Previous Reviewer 3)

As I can see, this is an extensively revised version of your manuscript. I consider that the manuscript should be published.

Author Response

Dear Reviewers,

               Thank you for your suggestions regarding the manuscript entitled “Metabolite profiling of the gut-renal-cerebral axis reveals a particular pattern in early diabetic kidney disease in T2DM patients”, by Lavinia Balint et al. Please find enclosed our point by point reply, as we have added the last detailes required.

Reviewer 1

               Thank you for your acceptance of our manuscript

Reviewer 2 Report (New Reviewer)

The study aims to find new gut-derived biomarkers implied in kidney and cerebrovascular endothelial dysfunction in DKD.

The Authors designed a cross-sectional study recruiting 90 DM2 patients and 20 healthy controls. DM2 patients were furtherly classified into three subgroups according to the albuminuria stage. No estimation of the sample size is declared, cases and controls heavily differ for age. The methods of metabolomic profiling are well described. The statistical analysis was adequate.

The results are clearly presented and the discussion is exhaustive.

The methodological limitations should be someway addressed, for example by adding in the title that this is a pilot study.

Author Response

Dear Reviewers,

               Thank you for your suggestions regarding the manuscript entitled “Metabolite profiling of the gut-renal-cerebral axis reveals a particular pattern in early diabetic kidney disease in T2DM patients”, by Lavinia Balint et al. Please find enclosed our point by point reply, as we have added the last detailes required.

Reviwer 2

               Thank you for your feedback. In order to be more specific concerning the sample size, we have reformulated the sentence in which the study design is described in subsection 4.1. Patients’ selection and Compliance with Ethical Standards:

Our research was based on a pilot study, that consisted of the collection of serum and urine samples from 110 eligible subjects (54 males and 56 females), of which, 90 were T2DM patients (group P) and 20 were healthy control subjects (group C), recruited from the records of general physicians.

               Concerning the age differences between controls and T2DM patients, we consider that they did not have an impact on our results, because controls were healthy subjects and they were corresponding with the inclusion criteria.

               In order to adress the methodological limitations, we have highlighted, in the paragraph regarding the study limitations, that our study is also a pilot one.

Lines 422 – 425: “First, this is a pilot and a cross-sectional study, therefore the results are observational and do not reflect a relation of causality.”

Reviewer 3 Report (New Reviewer)

In current study, authors used untargeted metabolomics approach using LC-MS to identify the gut derived metabolites in serum and urine from individuals with T2D. Authors focused on the identifying the gut derived metabolites and their involvement in podocyte injury and proximal tubular damage and endothelial dysfunction in kidney and cerebrovascular system and those metabolites role in pathogenesis of Diabetic kidney disease (DKD) in early stage. The presented clinical data generated from serum & urinary metabolomics approach is very intriguing and important contribution to the field of diabetes which allows understand progression of kidney disease including the cerebral vascular damage in early stage. The experimental design is appropriate and written well. However, authors need address several concerns before the current manuscript accepted for publication.

 Comments:

1)  Line 434, Table2 authors need to mentioned percentage of male and female individuals in each group. This can also allow if there are any gender differences in metabolites.

2)  In materials and methods, section 4.1 authors divided the group P into P1, P2, P3 according to uACR. However, uACR is similar in both P1 and control groups. How authors evaluated the group P1 as the patient demographs in Table 2 indicating that there are no significant or minimal differences or in all parameters compared to control groups? Explain.

3) As authors indicated that the changes in metabolites associated with tubular dysfunction, I would perform the ELISA assay for tubular injury marker KIM1 (kidney injury molecule). Authors need to present KIM1 results together with serum Lipid profile from these individuals.

4) In Results section, the Image resolution of Scores Plots in all the Figures were very poor. The image quality is very poor and hardly can fallow.  Need to fix.

5) As authors identified several gut derived metabolites related to nitrogen metabolic pathway and retinoic acid signaling pathway, Phenylalanine and tyrosine metabolism were changed in early stage of diabetes and may be associated with renal injury and impairment of BBB. Is there any evidence indicating that these metabolites treatment induce the podocyte, tubular and endothelial injury using invitro cell culture models? If there any such previous evidences have been showed, I would suggest to discuss in the discussion.

Author Response

Dear Reviewers,

               Thank you for your suggestions regarding the manuscript entitled “Metabolite profiling of the gut-renal-cerebral axis reveals a particular pattern in early diabetic kidney disease in T2DM patients”, by Lavinia Balint et al. Please find enclosed our point by point reply, as we have added the last detailes required.

Reviewer 3

               Thank you for your valuable remarks for improving our manuscript. They were taken under consideration and treated accordingly:

1)  Line 434, Table2 authors need to mentioned percentage of male and female individuals in each group. This can also allow if there are any gender differences in metabolites.

               The number of male and female in this study was almost equal, that was the reason we did not mention it in the table2. The study consisted of 110 subjects (54 males and 56 females). In the T2DM group the ratio between males/females was: 14/16 in P1, 13/17 in P2, 15/15 in P3. The control group consisted of 12 males an 8 females. In consequence, we do not consider that gender made any difference in the metabolite profiling. The table was redrafted accordingly

“The design of the study was based on the collection of serum and urine samples from 110 eligible subjects (54 males and 56 females), of which, 90 were T2DM patients (group P) and 20 were healthy control subjects (group C) recruited from the records of general physicians.”

Table 2. Demographic and clinical data

P1

P2

P3

C

Number of participants

30

30

30

20

Men (nr.,%)

14 (46.66%)

13 (43.33%)

15 (50%)

12 (60%)

Age (y)

68.41 ± 4.98

68.65 ± 4.91

68.84 ± 4.98

55.85 ± 7.25

DM duration (y)

9.6 ± 3.99

9.7 ± 3.99

12.78 ± 3.35

0

Serum creatinine (mg/dL)

0.82 ± 0.18

0.93 ± 0.21

1.07 ± 0.32

0.73 ± 0.08

eGFR (mL/min/1.73 m2)

90.42 ± 18.10

89.70 ± 18.19

77.85 ± 19.38

97.93 ± 11.71

UACR (mg/g)

7.38 ± 3.22

45.42 ± 57.08

319.86 ± 585.80

5 ± 0.23

HbA1c (%)

5 ± 0.23

6.42 ± 1.29

7.15 ± 1.60

4.98 ± 0.23

2)  In materials and methods, section 4.1 authors divided the group P into P1, P2, P3 according to uACR. However, uACR is similar in both P1 and control groups. How authors evaluated the group P1 as the patient demographs in Table 2 indicating that there are no significant or minimal differences or in all parameters compared to control groups? Explain.

               The patients in the P1 subgroup were evaluated by various criteria when compared with control group and other subgroups (P2,P3), besides uACR. Glucoregulation disbalances appear years before setting the diagnosis of DM and patients in P1 subgroup in our study have at least 5 years of DM duration based on inclusion criteria. This particular category of patients (the normoalbuminuric subgroup), by the time of the diagnosis of DM, already have proximal tubular dysfunction, before albuminuria becoming evident. In parallel, in our study, there were applied several other statistical analyses (untargeted univariate and semi-targeted analyses), after which the discrimination of metabolites between subgroups were made (P1 vs. C). Therefore, we consider that the discrimination between group C and P1 does not only rely on uACR classification, but also in DM  duration and the other analytical methods applied.

3) As authors indicated that the changes in metabolites associated with tubular dysfunction, I would perform the ELISA assay for tubular injury marker KIM1 (kidney injury molecule). Authors need to present KIM1 results together with serum Lipid profile from these individuals.

               The actual aim of the study was to observe the dynamics of metabolites based on untargeted multivariate and univariate analyses. We are aware of the fact that the our study is observational and cannot point out a relation of causality. The ELISA assay for KIM-1 overcomes the aim of our study, as we have not performed a targeted analysis, with subsequent metabolite quantification. A following study is under course, in which the targeted analysis of metabolites will be correlated with specific markers of renal damage (including KIM-1) and endothelial dysfunction.

4) In Results section, the Image resolution of Scores Plots in all the Figures were very poor. The image quality is very poor and hardly can fallow.  Need to fix.

               The images’ dimensions were increased, with respect to the IJMS template requirements, and the image resolution was enhanced for a better comprehension.

5) As authors identified several gut derived metabolites related to nitrogen metabolic pathway and retinoic acid signaling pathway, Phenylalanine and tyrosine metabolism were changed in early stage of diabetes and may be associated with renal injury and impairment of BBB. Is there any evidence indicating that these metabolites treatment induce the podocyte, tubular and endothelial injury using invitro cell culture models? If there any such previous evidences have been showed, I would suggest to discuss in the discussion.

               In our research, we have managed to point out a possible involvement in BBB regarding phenylalanine and tyrosine levels. Indeed, in literature there were some research studies focused on the correlation between metabolites and renal biopsies (Kumar Sharma et al. DOI: 10.1681/ASN.2013020126 ). Our subjects did not undergo renal biopsy and we do not posses renal biopsy tissues in order to correlate them with the metabolites found, to point out their involvement at a cellular level. For now, at this point of our untargeted metabolomic study there were not taken under consideration the correlation of metabolites with cell cultures, and implicit renal biopsies. Nevertheless, these aspects will be taken into consideration in the next, targeted stage.

Round 2

Reviewer 3 Report (New Reviewer)

Thank you. The authors addressed all the comments and responses were accepted. 

This manuscript is a resubmission of an earlier submission. The following is a list of the peer review reports and author responses from that submission.

Round 1

Reviewer 1 Report

I have read with great interest the manuscript written by the authors. The manuscript is well written although there are several concerns. Please find the following comments for consideration.

・Is it able to interpret that the difference observed in the comparison between P and C group is derived from the presence of diabetes? More than half of the patients with P group have albuminuria. Is there a possibility that the difference is derived from albuminuria? In addition, we can’t exclude the influence of anti-diabetes drugs. I would like to know the medication in each group.

・Can you exclude the influence of glycemic control status about the change in comparison of P1,P2, and P3 groups? Patients with P3 group have not only higher levels of albuminuria but also has higher HbA1c.

・Even though the metabolites that showed differences in this comparison have been reported to be associated with endothelial damage, I think it is not enough to discuss the association between the metabolites and brain. It would be better to show the clinical study that discuss the association some cerebral marker and metabolites.

・The VIP score, MDA score, and AUC curve were used to select potential biomarkers, but it was not clear how the final selection was made from the three analyses.

・The issue about tryptophan was not mentioned in results including figure and table, but the authors discussed in the discussion. I would like to know the results related to tryptophan for more detail.

・The serum levels of p-cresyl sulfate and indoxyl sulfate were increased, on the other hand urinary level of p-cresyl sulfate was decreased but indoxyl sulfate was increased. How do the authors consider about the different pattern of p-cresyl sulfate and indoxyl sulfate in serum and urine unless these metabolites are resemble in many aspects? Is it come from the difference of etiology of excrete?

・How do the authors assume that these changes contribute to the pathogenesis of early DKD?

・I think you should add the “)” after “214.2550 (Indoxyl sulfate” in 151 line.

・The explanation of Table1a may end the middle of sentence. The last word of the sentence is “also”. 

Reviewer 2 Report

This review discussed the role of the CD4 T cells in IBD, including Th1, Th2, Th9, Th17, Th22, Tfh, and Treg cells. They further highlighted how these cells adapt to the environment and interact with other cell populations to promote or inhibit the development of IBD. This is a comprehensive review of the relationships between adaptive immune responses and IBD, which could provide some new insights for the treatment or prevention of IBD. I have some minor comments to improve this manuscript.

Th2 cells are involved in the elimination of extracellular microbes and parasites and emerging data showed that Th2 cells play an important role in promoting the development of IBD, such as Blastocystis. Recent data showed this parasite colonization increased the accumulation of Th2 and Tregs, which promotes faster recovery from experimental-caused colitis. 

Metabolites produced by gut microbiota could induce Treg cells differentiation, such as bile acids, SCFAs, etc. It would be better to state the role of those small molecules in regulating the Treg cells and then affect the development of IBD.

Reviewer 3 Report

14 authors have written an article of 8.5 pages (in MDPI style, covering 2/3 of a page) regarding Metabolite profiling of the gut-renal-cerebral axis reveals a particular pattern in early diabetic kidney disease in T2DM patientsThe topic is not a new one. No novelty was revealed, no specific aspects have been described in the aim of the study or in other parts, etc. Detailing, please see my suggestions:

Shape suggestions

Please insert the corresponding author’s email at Correspondence email.

Please check the Instructions for authors related to the text setting (Abstract: not bolded, Palatino Linotype 9; Subtitles, not bolded but in Italics), etc., and apply them.

Liter is the international unit of measure for volume, having as symbol L. So, you must use mL, dL, etc. Please check and correct in the entire manuscript, including in tables/figures. (i.e. L383, Table 2.)

Numerical values must be written in English style. Table 2. 1,73 must be 1.73. Check and revise the entire manuscript.

Unit of measure for BMI Table 2. (kg/m2), please insert 2 (for square meter) at superscript.

L393. The link must be inserted as reference, in square brackets, and inserted also in the References section, where “Accesesd on day/month/year” must be also added. Check the Instructions for authors.

Reduce the self citations in the manuscript (i.e. Gadalean).

Content suggestions

Results

It is not clear how the patients included in the research were selected, both in the study group and in the control group.

How was the sample size calculated?

How do you explain the equality ratio regarding the number of patients from the 3 subgroups in the study group? If you chose an equal number, you must mention the criteria on the basis of which you did so.

The demographic and clinical characteristics of the subjects included in the study were presented in Table 2. There is no reference in the article to these characteristics and no reference to Table 2. What is their relevance for the study?

The average age seems to be significantly lower in the control group. Please detail.

Statistics is irrelevant, as 33 is the minimum number of patients/ EACH group, for having a relevant statistic (which anyone with minimal knowledge of statistics knows).

Discussion

Should be improved. Please discuss the role of probiotic supplementation in order to ameliorate the gut microbiota and therefore improve the aminoacidic profile of patients with early type 2 diabetes mellitus; I suggest checking and referring to PMID: 33802777; PMID: 33662428

Also detail the implications of glp-1 therapy that can impact all the analyzed axis in clinical practice. I suggest PMID: 32765722.

As well, you should discuss the role of metabolic profile in the personalized care of diabetes patients.